# COVID-19 Vaccine Hesitancy in a Population-Based Study of Minnesota Residents

**DOI:** 10.3390/vaccines11040766

**Published:** 2023-03-30

**Authors:** Sallee Brandt, Ryan T. Demmer, Sara Walsh, John F. Mulcahy, Evelyn Zepeda, Stephanie Yendell, Craig Hedberg, Angela K. Ulrich, Timothy Beebe

**Affiliations:** 1Division of Epidemiology and Community Health, School of Public Health, University of Minnesota, Minneapolis, MN 55455, USA; 2Health Sciences, NORC at the University of Chicago, Chicago, IL 60603, USA; 3Division of Health Policy and Management, School of Public Health, University of Minnesota, Minneapolis, MN 55455, USA; 4Minnesota Department of Health, St. Paul, MN 55164, USA; 5Division of Environmental Health Sciences, School of Public Health, University of Minnesota, Minneapolis, MN 55455, USA; 6Center for Infectious Disease Research and Policy, University of Minnesota, Minneapolis, MN 55455, USA

**Keywords:** vaccine hesitancy, booster vaccine, infectious disease, public health, surveillance, public health preparedness, infections

## Abstract

COVID-19 continues to be a public health concern in the United States. Although safe and effective vaccines have been developed, a significant proportion of the US population has not received a COVID-19 vaccine. This cross-sectional study aimed to describe the demographics and behaviors of Minnesota adults who have not received the primary series of the COVID-19 vaccine, or the booster shot using data from the Minnesota COVID-19 Antibody Study (MCAS) collected through a population-based sample between September and December 2021. Data were collected using a web-based survey sent to individuals that responded to a similar survey in 2020 and their adult household members. The sample was 51% female and 86% White/Non-Hispanic. A total of 9% of vaccine-eligible participants had not received the primary series and 23% of those eligible to receive a booster had not received it. Older age, higher education, better self-reported health, $75,000 to $100,000 annual household income, mask-wearing, and social distancing were associated with lower odds of hesitancy. Gender, race, and previous COVID-19 infection were not associated with hesitancy. The most frequently reported reason for not receiving a COVID-19 vaccination was safety concerns. Mask-wearing and being age 65 or older were the only strong predictors of lower odds of vaccine hesitancy for both the primary series and booster analyses.

## 1. Introduction

Infections caused by SARS-CoV-2, referred to as COVID-19, continue to be a significant public health concern in the United States. Vaccinations have been developed and approved to reduce the risk of infection, severe illness, and death from COVID-19 [1,2]. Despite being safe, highly effective, free to the public, and widely available, a notable portion of individuals is not vaccinated. Approximately one-third of the eligible United States population has not received the complete primary vaccination series and around half of those that are fully vaccinated have not received a third dose as of November 2022 [3]. Suboptimal vaccine uptake appears to be continuing, with just over 15 percent of Americans having received an updated, bivalent booster vaccine as of January 2023 [3].

Vaccine hesitancy has previously been defined as “…a state of indecision and uncertainty that precedes a decision to become (or not become) vaccinated” [4], and the specific nature of hesitancy varies by vaccine, geography, time, and other contexts [4]. Vaccine hesitancy is thought to be a major driver of poor vaccine uptake in the United States [4]. The previous literature suggests that hesitancy in receiving COVID-19 vaccinations is higher among populations with fewer years of education [5,6,7]. An association between vaccine hesitancy and living in a region with a higher proportion of Republican voters has also been described [5,7]. Commonly reported reasons for being hesitant to receive a COVID-19 vaccine include concerns about side effects [6,8,9,10] and not trusting the vaccine [6,8,11,12]. It is important to understand the reasons for vaccine hesitancy to inform public health efforts to increase vaccination rates and reduce COVID-19-related hospitalizations and mortality.

The aim of the study was to investigate the relationship between demographic measures, prior COVID-19 infection, and a hesitancy to receive the primary series or a booster of the COVID-19 vaccine among Minnesota residents enrolled in the Minnesota COVID-19 Antibody Study (MCAS).

## 2. Materials and Methods

The Minnesota COVID-19 Antibody Study (MCAS) drew its respondents from the COVID-19 Impact Survey (CIS), which was a population-based, cross-sectional survey that gathered information on respondents’ mental and physical health, and economic security from U.S households in 10 states and 8 Metropolitan Statistical Areas enrolled over three waves of data collection between April and June of 2020. CIS data were collected using the AmeriSpeak Panel^®^, The National Opinion Research Center (NORC) at the University of Chicago’s probability-based panel designed to be representative of the U.S. household population. Data for the regional estimates, such as Minnesota, were collected using a multi-mode address-based sampling approach that allows residents of each area to complete the interview via the web or with a NORC telephone interviewer.

MCAS recruited CIS respondents residing in Minnesota, originally recruited to CIS using address-based sampling, who had given their consent to be contacted for future studies. MCAS respondents were contacted for two rounds: Round One ran from December 2020 to February 2021, and Round Two ran from September to December 2021. This analysis includes data collected in Round Two, which incorporates previously sampled individuals from the first round of data collection in 2020. Round Two expanded the original MCAS sample (defined as the starting sample provided by CIS minus any respondents who withdrew) by rostering the household of original MCAS respondents, also known as Panel One respondents, and enrolling eligible household members (age 18 and older) to participate in the survey data collection and antibody blood testing. New respondents were designated Panel Two respondents. Children of original MCAS respondents, ages 6 through 17, were eligible to participate in the antibody blood testing but were not asked to complete a survey. Survey data were collected via a web-based survey.

The MCAS collected the first round of data in 2020 (N = 907). This study includes a subset (n = 770) of respondents who participated in a second round of surveys administered between September and December 2021 and had demographic and exposure information available. Only individuals that were eligible to receive a booster (i.e., they had received a primary series vaccine) were asked booster vaccine questions (n = 719), and of those, n = 714 responded to the booster vaccine questions and are included in this analysis.

Basic demographic information was self-reported by respondents and included age, gender, race/ethnicity, household income, number of household members, and education level. For analysis, age was categorized into five groups (18–34, 35–44, 45–54, 55–64, 65+), and race/ethnicity was analyzed as a binary variable (White/Non-Hispanic vs. Non-White). Individuals included in the first round of data collection in 2020 were not asked age, gender, race/ethnicity, or education level but were asked those questions in the second round. Household income, number of household members, and education level were coded as categorical variables with the lowest category as the reference. Data on perceived current general health, COVID-19 mitigation behaviors, and previous COVID-19 infection were self-reported. General health status was self-reported and categorized as excellent/very good or good/fair/poor. Individuals were asked if they use masks or social distancing in response to the COVID-19 pandemic. A previous COVID-19 infection was defined by asking if the respondent had ever been told by a healthcare professional that they had COVID-19. A summary of respondent characteristics is presented in Table 1.

The primary outcome in this analysis was vaccine hesitancy, defined as being eligible for the COVID-19 vaccine primary series or booster dose(s), but had not yet received it. Respondents were asked if they have received a COVID-19 vaccine or if they had an appointment to receive one. Individuals that reported not receiving a COVID-19 vaccine and responded that they did not have an appointment to receive a COVID-19 vaccine were considered vaccine hesitant toward the primary series. Further information was collected on reasons for not undergoing vaccination. Individuals were given a list of reasons and could select more than one option. There was also a write-in option that allowed individuals to describe their specific reasons for not receiving a COVID-19 vaccine. Reasons that were written in were not included in this analysis. For those that received a COVID-19 vaccination, booster shot intentions were recorded. Respondents could answer that they will get it, already received it, will not receive it, or are not sure about receiving a booster shot. Individuals who reported they will not receive a booster shot or were “unsure” were considered booster vaccine-hesitant. Reasons for not receiving the third dose of the COVID-19 vaccine were written in and were not categorized or analyzed in this study.

Statistical analysis was conducted using SAS OnDemand. Descriptive statistics were used to describe the prevalence of the exposures among the study sample (Table 1). Crude and adjusted logistic regression models were used to analyze the odds of vaccine hesitancy for the primary series of COVID-19 vaccination and the booster vaccine. The adjusted model contained age group, gender, race/ethnicity, household income bracket, household size, educational attainment, general health, mask-wearing, social distancing, and self-reported prior COVID-19 infection. The frequency of use for the reasons for primary series vaccine hesitancy was recorded. An alpha of 0.05 was used to determine the statistical significance of two-sided confidence intervals. All frequency estimates and regression models applied sampling weights to make our sample representative of the adult Minnesota population.

## 3. Results

Study sample characteristics are presented in Table 1. The sample population was 50.9% female and 85.9% White/Non-Hispanic, and around 50% of the participants were 55 years of age or older (46.5%). A total of 45.2% of the sample population had a household income above $75,000. Of the respondents, 20.7% lived alone and 8.5% lived in a household with three or more other people. Almost 39% of the sample population had a bachelor’s degree or a higher level of education. A total of 72% self-reported their general health status as “very good” or “excellent”. For COVID-19 mitigation factors, 92% of the sample population reported engaging in mask-wearing and 67.6% reported practicing social distancing. Of the respondents, 89.5% had not been told by a health professional that they tested positive for COVID-19. A total of 51 participants (9.3%) were hesitant to receive the primary series of the COVID-19 vaccine. Additionally, 130 participants (23.0%) who received a primary-series COVID-19 vaccine had not yet received a booster although they were eligible.

In the crude logistic regression models for primary series vaccination (Table 2), individuals that were 55–64 years old (OR: 0.19; 95% CI: 0.04–0.98) and 65 years or older (OR: 0.02; 95% CI: 0.002–0.18) had significantly lower odds of vaccine hesitancy compared to those 18–34 years old in the fully adjusted model. The number of household members was also significantly associated with primary series hesitancy. Those with four or more household members had significantly higher odds of vaccine hesitancy compared to those with no household members in the crude model (OR: 5.81; 95% CI: 1.36–24.70). An annual household income of $75,000 to $100,000 was associated with lower odds of vaccine hesitancy relative to household incomes of less than $25,000 in both the unadjusted (OR: 0.10; 95% CI 0.02–0.45) and adjusted models (OR: 0.06; 95% CI 0.002–0.50); however, no other income categories were significantly associated with hesitancy. Educational attainment was not significantly associated with primary vaccine hesitancy, except among people with professional or doctorate degrees (n = 58), where all sampled individuals were vaccinated. Self-reported health status of excellent or good was associated with lower odds of vaccine hesitancy in the adjusted model (OR: 0.38; 95% CI 0.16–0.91). Mask-wearing (OR: 0.22; 95% CI: 0.08–0.65) was significantly associated with lower odds of vaccine hesitancy in the crude model. Neither mask-wearing (OR: 0.19; 95% CI: 0.03–1.04) nor social distancing (OR: 1.07; 95% CI 0.31–3.68) were significant in the adjusted model. Gender, race/ethnicity, and previous COVID-19 infection were not significantly associated with primary-series vaccine hesitancy in the crude or adjusted models.

In the booster hesitancy analysis (Table 2), those 65 years or older had significantly lower odds of booster hesitancy in the crude (OR: 0.13; 95% CI: 0.04–0.36) and the adjusted (OR: 0.08; 95% CI: 0.03–0.26) models. Those aged 55–64 also had significantly lower odds of booster hesitancy compared to those aged 13–34 in the adjusted model (OR: 0.37; 95% CI: 0.15–0.95). There was no significant association between the number of household members and booster vaccine hesitancy in the crude or adjusted model. Those with a bachelor’s degree or master’s degree had significantly lower odds of booster hesitancy compared to those with high school diplomas or less education in the crude (bachelor’s OR: 0.29; 95% CI: 0.12–0.67; master’s OR: 0.31; 95% CI 0.13–0.72) and the adjusted model (bachelor’s OR: 0.20; 95% CI: 0.07–0.56; master’s OR: 0.17; 95% CI: 0.05–0.56). Individuals with a professional or doctorate degree had significantly lower odds of booster hesitancy in the crude model (OR: 0.20; 95% CI: 0.04–0.97), but not in the adjusted model. Mask-wearing and social distancing were significantly associated with lower odds of booster hesitancy in the crude (mask OR: 0.07; 95% CI: 0.02–0.21; social distancing OR: 0.29; 95% CI: 0.15–0.58) and adjusted (mask OR: 0.12; 95% CI: 0.02–0.58; social distancing OR: 0.39; 95% CI: 0.18–0.81) models. Gender, race/ethnicity, household size, household income, self-reported health status, and previous COVID-19 infection were not significantly associated with booster hesitancy in the crude or adjusted models.

Among respondents who had not yet completed the COVID-19 primary-series vaccine, the most reported reason for not doing so was concerns about the safety of the vaccines (75.4%). Those who indicated safety as a concern were asked a further question with multiple options to narrow down the specific safety concern about the vaccines (Figure 1). Among those who indicated that safety was a concern, the top specific concerns within the safety category were about side effects (86.8%), concerns that the vaccine was developed too quickly (78.9%), and not trusting the government (55.3%) (specific reasons not displayed in Figure 1). One-quarter (27.2%) of unvaccinated respondents reported that the reason they had not received a COVID-19 vaccine was because they had previously been diagnosed with COVID-19. No respondent indicated that they were unsure about costs or how to get the vaccine.

## 4. Discussion

The state of Minnesota experienced rates of primary vaccine hesitancy that were notably lower than the broader United States and a select global sample (9.3% vs. 19.8% and 20.9%, respectively) [13]. A recent literature review found that vaccine hesitancy varied from around 10% to almost 90% within the United States by individual states and demographic groups, and our findings suggest Minnesota has lower hesitancy rates than many other states [14]. However, Minnesota saw higher unadjusted rates of booster vaccine hesitancy compared to those same populations (23.0% vs. 13.0% and 12.1%, respectively) [13]. Older age, household income of $75,000 to $100,000, and excellent or very good self-reported health were significantly associated with lower odds of primary-series vaccine hesitancy compared to their respective reference groups in the adjusted analysis. Four or more household members were associated with higher odds of primary-series vaccine hesitancy in the adjusted model. In the adjusted booster hesitancy models, being 55 years or older, higher educational attainment, mask-wearing, and social distancing were significantly associated with decreased odds of booster hesitancy compared to the reference groups. Gender and race/ethnicity were not associated with primary series or booster hesitancy in any model. The most reported reason for vaccine hesitancy was around the safety of the vaccine.

These results generally align with previous literature analyzing population-based surveys on COVID-19 vaccination. There is evidence to support younger individuals having higher rates of hesitancy compared to older individuals [7,15,16]. There was not a statistically significant association between sex or race found in previous studies on this topic, but the sample in this analysis was not as racially or ethnically diverse as other samples with over 90% of individuals identifying as White/Non-Hispanic [5,7,17]. The most frequently reported reason for primary-series vaccine hesitancy was concerns about the safety of the vaccine which is consistent with previous literature [10,15,18]. Previous research has found possessing a bachelor’s degree or higher to be associated with lower odds of primary vaccine hesitancy [5,7,15,16], but unassociated with booster vaccine hesitancy [15]; however, our models showed no significant relationship between education and primary hesitancy or with booster vaccine hesitancy. It should be noted that our model may have been under-powered to detect a significant relationship between education and primary hesitancy with only 51 individuals reporting primary vaccine hesitancy. Excellent or very good health was negatively associated with primary vaccine hesitancy but positively associated with booster vaccine hesitancy. This is consistent with findings from a recent study where excellent or very good health was associated with booster vaccine hesitancy among fully vaccinated nurses in Greece [19].

Our models found there were different demographic factors and behaviors associated with primary vaccine hesitancy relative to booster vaccine hesitancy. This suggests there may be differences in the underlying reasons for hesitancy towards initial and subsequent vaccinations. Though our data did not allow us to examine reported reasons for booster hesitancy, we were able to compare respondents’ reported reasons for primary vaccine hesitancy to other estimates in the literature. In total, 75.4% of respondents reported safety as a reason for primary vaccine hesitancy, and other research has shown safety to be a common concern for vaccine hesitancy [4,6,8,9,20,21]; however, only 32% of booster-hesitant respondents reported safety as a reason for hesitancy in a sample of people Aged 60 Years and Older in China [22]. The lower prevalence of safety as a concern among the booster vaccine hesitant may explain why good/fair/poor health was protective of primary vaccine hesitancy but not booster vaccine hesitancy in our models.

The primary strength of this study is that it is a population-based survey. There were data collected on many different variables which allowed us to include many possible confounders in the adjusted models. One limitation of this study is the lack of racial diversity among the participants. Racial inequities are apparent in the health outcomes of COVID-19 and further research should be conducted to understand the impacts of structural racism on COVID-19 vaccine uptake. Another limitation of this dataset is that it did not collect information about political leanings or media sources as a possible confounder, or geographic and community measures that may be associated with these confounders. Previous literature has identified political affiliation to be associated with vaccine hesitancy [5,7,15]. A third limitation of this analysis relates to the reasons for vaccine hesitancy. Reasons for hesitancy were collected in categorical options and a textbox where respondents could write in reasons that were not listed for the primary vaccine hesitancy. In contrast, reasons for booster hesitancy were only collected in a textbox format. Written-in responses were not analyzed. The MCAS also allowed for multiple members of the same household to be sampled, and we did not cluster observations by household. Finally, we used the second round of MCAS for our data analysis, which only sampled respondents who participated in the first wave of the study. By only sampling from people that consented to a second wave of data collection, we may be introducing selection bias that makes our sample less representative of Minnesota as a whole.

## 5. Conclusions

Being 55 years or older was the only significant exposure in the crude and adjusted models for both the primary series and the booster analyses. Understanding the relationship between younger age and hesitancy is important to inform future public health efforts to increase vaccination rates. The differential effect of educational attainment on primary and booster vaccine hesitancy should also be further explored. Further research should also investigate media sources and their role in hesitancy as a possible confounder between age, education, income, health, and vaccine hesitancy.

## Figures and Tables

**Figure 1 vaccines-11-00766-f001:**
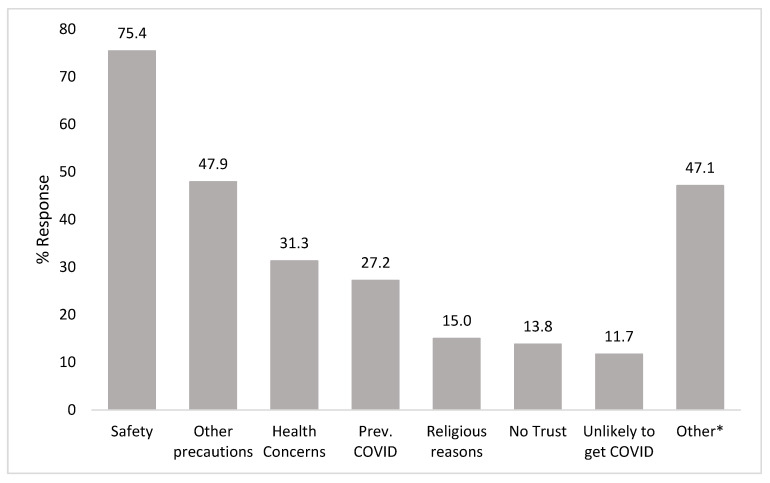
Reasons for primary-series vaccine hesitancy. * “Other” category includes write-in responses (41.4%), “I want to wait and see until more people get the vaccine” (29.1%), “My friends and/or family are not getting it” (0.7%), and “I’m concerned about what other people might think” (0.4%).

**Table 1 vaccines-11-00766-t001:** Minnesota COVID-19 Antibody Study (MCAS) Respondent Characteristics.

	Total		Primary Vaccine Hesitant		Booster Vaccine Hesitant	
Variable	Frequency (Weighted %)	*p*-Value *	Frequency (Weighted %)	*p*-Value *	Frequency (Weighted %)	*p*-Value *
Total Sample Size	770		51		130	
Age		<0.0001		0.0063		0.0018
18–34	156 (28.3%)		17 (48.0%)		33 (38.9%)	
35–44	124 (12.9%)		13 (23.4%)		30 (16.1%)	
45–54	122 (12.3%)		11 (14.4%)		24 (16.3%)	
55–64	160 (21.0%)		6 (13.1%)		28 (21.3%)	
65+	208 (25.5%)		4 (1.5%)		15 (7.4%)	
Gender		0.77		0.86		0.21
Male	328 (49.1%)		25 (47.3%)		47 (41.0%)	
Female	442 (50.9%)		26 (52.7%)		83 (59.0%)	
Race/Ethnicity		<0.0001		0.63		0.12
White/Non-Hispanic	706 (85.9%)		47 (81.7%)		114 (78.9%)	
Non-White	64 (14.1%)		4 (18.3)		16 (21.1%)	
Household Income		<0.0001		0.38		0.79
Under $25,000	96 (12.1%)		7 (21.1%)		17 (7.2%)	
$25,000 to under $50,000	127 (26.5%)		11 (32.1%)		23 (33.3%)	
$50,000 to under $75,000	110 (16.2%)		5 (14.8%)		19 (14.8%)	
$75,000 to under $100,000	151 (16.3%)		7 (3.4%)		28 (16.9%)	
$100,000 to under $150,000	157 (16.9%)		11 (14.4%)		24 (17.1%)	
$150,000 or more	129 (12.0%)		10 (14.3%)		19 (10.8%)	
Number of		<0.0001		0.024		0.089
Household Members						
0	149 (20.7%)		4 (14.6%)		21 (13.8%)	
1	335 (41.1%)		16 (31.0%)		51 (30.7%)	
2–3	224 (29.7%)		16 (28.0%)		44 (43.8%)	
4+	62 (8.5%)		15 (26.6%)		14 (7.1%)	
Highest Education Level		<0.0001		NA **		0.019
High School Graduate or less	81 (28.6%)		10 (32.0%)		18 (42.4%)	
Some college, no degree	100 (17.5%)		12 (21.0%)		18 (18.8%)	
Associate degree	82 (14.9%)		8 (15.4%)		15 (16.8%)	
Bachelor’s degree	280 (21.7%)		14 (20.2%)		46 (12.4%)	
Master’s degree	169 (12.5%)		7 (11.4%)		29 (7.5%)	
Professional/Doctorate degree	58 (4.7%)		0 (0%)		4 (2.1%)	
General Health Status		<0.0001		0.11		0.12
Good/Fair/Poor	184 (28.2%)		17 (43.7%)		30 (19.2%)	
Excellent/Very Good	586 (71.8%)		34 (56.3%)		100 (80.8%)	
Reported Mask-Wearing		<0.0001		0.0035		<0.0001
No	52 (8.0%)		15 (23.5%)		26 (21.6%)	
Yes	718 (92.0%)		36 (76.5%)		104 (78.5%)	
Reported Social Distancing		<0.0001		0.60		0.0003
No	238 (32.4%)		29 (37.3%)		67 (53.7%)	
Yes	532 (67.6%)		22 (62.7%)		63 (46.3%)	
Previous Self-Reported		<0.0001		0.20		0.30
COVID-19 Infection						
Yes	80 (10.5%)		15 (17.3%)		18 (7.3%)	
No/Not sure	690 (89.5%)		36 (82.7%)		112 (92.7%)	

* *p*-values obtained using Chi-Square test. ** Weighted Chi-Square test cannot be calculated for cells with 0 observations.

**Table 2 vaccines-11-00766-t002:** Unadjusted and Adjusted ORs for primary-series vaccine hesitancy and booster vaccine hesitancy.

	Primary-Series Hesitancy (N = 770)		Booster Vaccine Hesitancy (N = 714)	
	Unadjusted OR (95% CI)	Adjusted OR (95% CI)	Unadjusted OR (95% CI)	Adjusted OR (95% CI)
Age				
18–34	Ref	Ref	Ref	Ref
35–44	1.09 (0.34, 3.43)	1.40 (0.30, 6.52)	0.88 (0.31, 2.50)	0.85 (0.29, 2.53)
45–54	0.63 (0.16, 2.53)	0.40 (0.08, 2.00)	0.90 (0.33, 2.50)	0.83 (0.30, 2.30)
55–64	0.33 (0.08, 1.38)	**0.19 (0.04, 0.98) ***	0.56 (0.23, 1.40)	**0.37 (0.15, 0.95) ***
65+	**0.03 (0.005, 0.18) ***	**0.02 (0.002, 0.18) ***	**0.13 (0.04, 0.36) ***	**0.08 (0.03, 0.26) ***
Gender				
Male	Ref	Ref	Ref	Ref
Female	1.09 (0.43, 2.73)	1.00 (0.36, 2.81)	1.55 (0.78, 3.10)	1.64 (0.81, 3.31)
Race/Ethnicity				
White/Non-Hispanic	Ref	Ref	Ref	Ref
Non-White	1.41 (0.35, 5.80)	0.70 (0.13, 3.89)	2.08 (0.81, 5.34)	1.47 (0.53, 4.05)
Household Income				
Under $25,000	Ref	Ref	Ref	Ref
$25,000 to under $50,000	0.66 (0.15, 2.86)	0.80 (0.17, 3.87)	2.41 (0.83, 7.00)	2.66 (0.90, 7.87)
$50,000 to under $75,000	0.48 (0.09, 2.65)	0.56 (0.09, 3.38)	1.51 (0.45, 5.10)	2.97 (0.97, 9.07)
$75,000 to under $100,000	**0.10 (0.02, 0.45) ***	**0.06 (0.006, 0.50) ***	1.63 (0.51, 5.27)	2.78 (0.96, 8.04)
$100,000 to under $150,000	0.44 (0.09, 2.24)	0.48 (0.06, 3.59)	1.70 (0.53, 5.48)	1.95 (0.61, 6.25)
$150,000 or more	0.65 (0.15, 2.75)	0.46 (0.06, 3.61)	1.56 (0.47, 5.17)	2.66 (0.67, 10.59)
Number of				
Household Members				
0	Ref	Ref	Ref	Ref
1	1.07 (0.27, 4.36)	1.91 (0.55, 6.62)	0.81 (0.31, 2.13)	1.23 (0.51, 2.93)
2–3	1.37 (0.31, 5.96)	0.54 (0.14, 2.15)	2.03 (0.74, 5.56)	1.63 (0.66, 4.01)
4+	5.81 (1.36, 24.70) *	3.25 (0.61, 17.30) *	1.30 (0.35, 4.87)	0.56 (0.12, 2.60)
Highest Education Level				
High School Graduate or less	Ref	Ref	Ref	Ref
Some college, no degree	1.08 (0.27, 4.36)	0.63 (0.14, 2.77)	0.65 (0.25, 1.72)	0.58 (0.19, 1.75)
Associate degree	0.91 (0.22, 3.71)	1.45 (0.40, 5.19)	0.66 (0.22, 1.97)	0.50 (0.14, 1.74)
Bachelor’s degree	0.81 (0.21, 3.19)	0.68 (0.14, 3.28)	0.29 (0.12, 0.67) *	0.20 (0.07, 0.56) *
Master’s degree	0.79 (0.18, 3.49)	0.80 (0.09, 7.48)	0.31 (0.13, 0.72) *	0.17 (0.05, 0.56) *
Professional/Doctorate degree	Undefined **	Undefined **	0.20 (0.04, 0.97) *	0.21 (0.03, 1.31)
General Health Status				
Good/Fair/Poor	Ref	Ref	Ref	Ref
Excellent/Very Good	0.47 (0.18, 1.23)	**0.38 (0.16, 0.91) ***	1.72 (0.85, 3.51)	1.79 (0.83, 3.89)
Reported Mask-Wearing				
No	Ref	Ref	Ref	Ref
Yes	**0.22 (0.08, 0.65) ***	0.19 (0.03, 1.04)	**0.07 (0.02, 0.21) ***	**0.12 (0.02, 0.58) ***
Reported Social Distancing				
No	Ref	Ref	Ref	Ref
Yes	0.79 (0.32, 1.93)	1.07 (0.31, 3.68)	**0.29 (0.15, 0.58) ***	**0.39 (0.18, 0.81) ***
Previous Self-Reported COVID-19 Infection				
Yes	Ref	Ref	Ref	Ref
No/Not sure	0.52 (0.19, 1.44)	0.44 (0.14, 1.41)	1.51 (0.68, 3.35)	1.06 (0.41, 2.73)

* *p* < 0.05. ** OR and 95% confidence interval unable to be determined as all individuals reporting having a professional or doctorate degree received at least one vaccine.

## Data Availability

Data will not be publicly available due to privacy/data use agreements.

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
