# Peer review of "COVID-19 Vaccine Hesitancy in a Population-Based Study of Minnesota Residents"

_vaccines, 2023, doi:10.3390/vaccines11040766_

Round 1

Reviewer 1 Report

The study objective to investigate the factor-related willingness to COVID-19 vaccination was well processed and the manuscript is very well written even if this hesitancy in population was investigated by many authors across the world. I suggest a few minor comments.

1) Explanation of the abbreviations: NORC (page 2 line 66), ABS (page 2, line 69)

2) Missing bracket (page 2, line 84)

3) Page 3, line 123-124:

"The adjusted model contained every covariable." - it should be better to report all covariable (such as age, sex, race/ethnicity etc.)

4) Page 3, line 125-126

"A p-value less than or equal to 0.05 was considered significant." - It should be rewritten using level of significance alpha including two- or one-side confidence interval. 

5) Table 1 - it should be placed in chapter "Results".

Author Response

Comments and Suggestions for Authors:

The study objective to investigate the factor-related willingness to COVID-19 vaccination was well processed and the manuscript is very well written even if this hesitancy in population was investigated by many authors across the world. I suggest a few minor comments.

1) Explanation of the abbreviations: NORC (page 2 line 66), ABS (page 2, line 69)

            Both abbreviations are now accompanied by explanations

2) Missing bracket (page 2, line 84)

A bracket has been added on page 2, line 84.

3) Page 3, line 123-124:

"The adjusted model contained every covariable." - it should be better to report all covariable (such as age, sex, race/ethnicity etc.)

            Thank you for this suggestion. All covariates in the multivariate are now listed on on page 3, lines 123-126 :

“The adjusted model contained age group, gender, race/ ethnicity, household income bracket, household size, educational attainment, general health, mask wearing, social distancing, and self-reported prior COVID-19 infection.”

4) Page 3, line 125-126

"A p-value less than or equal to 0.05 was considered significant." - It should be rewritten using level of significance alpha including two- or one-side confidence interval. 

            We now explain how statistical significance was determined using the following sentence on page 3, lines 127 -128.

"An alpha of 0.05 was used to determine statistical significance of two-sided confidence intervals"

5) Table 1 - it should be placed in chapter "Results".

            Table 1 now begins after the first paragraph in our results section, on page 3, line 140.

Reviewer 2 Report

Brandt and colleagues explore the reason of vaccine hesitancy in Minnesota, reporting that older age was the main driver predisposing individuals to get vaccinated and receive a booster dose.

Serval reports  have already addressed these points, still the work has the merit to describe specific situation (i.e. linked to state of Minnesota) and explore the differences between hesitancy towards the primary vaccination course and the booster dose.

However, the sample size is rather limited if considering that the title refers to “Minnesota residents”

Major concerns:

1) there are some major differences between the factors associated with primary and booster vaccine hesitancy…the most striking: OR direction is even opposite in case of health status and income. Why? I strongly encourage the authors to find reasons, as this may rally provide a novel result and improve the originality of the work (a qualitative sub study may help).

2) Fig 1 could be stratified according to vaccine dose (primary or booster)

3) the literature in this topic is wide, authors should include and discuss a larger part of it

4) the study is based on the follow-up of a previous cohort, and only subjects who had given their consent to be contacted, have been recruited. This can constitute a selection bias and should be discussed

Minor concerns:

5) a more direct comparison with favors associated with vaccine hesitancy other US states with different demographic and social characteristics would be important to highlight the findings of the study

6) Data presented at lines 175-177 are not represented in any figure

7) no description of living conditions /e.g. rural/urban) is reported

Author Response

Major concerns:

1) there are some major differences between the factors associated with primary and booster vaccine hesitancy…the most striking: OR direction is even opposite in case of health status and income. Why? I strongly encourage the authors to find reasons, as this may rally provide a novel result and improve the originality of the work (a qualitative sub study may help).

            Thank you for highlighting the importance of these findings. Though our data does not allow us to provide a causal explanation for these differences, we have added a paragraph to our discussion that emphasizes these findings and provides possible explanations on page 8, lines 224 – 239.

“.Excellent or very good health was negatively associated with primary vaccine hesitancy, but positively associated with booster vaccine hesitancy. This is consistent with findings from a recent study where excellent or very good health was associated with booster vaccine hesitancy among fully vaccinated nurses in Greece [18]. Our models found there were different demographic factors and behaviors associated primary vaccine hesitancy relative to booster vaccine hesitancy. This suggests there may be differences in the underlying reasons for hesitancy towards initial and subsequent vaccinations. Though our data did not allow us to examine reported rea-sons for booster hesitancy, we were able to compare respondents’ reported reasons for primary vaccine hesitancy to other estimates in the literature. 75.4% of respondents reported safety as a reason for primary vaccine hesitancy, and other research has shown safety to be a common concern for vaccine hesitancy [4,6,8,9,19,20]; however, only 32% of booster hesitant respondents reported safety as a reason for hesitancy in a sample of people Aged 60 Years and Older in China [21]. The lower prevalence of safety as a concern among booster vaccine hesitant may explain why we good/fair/poor health was protective of primary vaccine hesitancy, but not booster vaccine hesitancy in our models.”     

2) Fig 1 could be stratified according to vaccine dose (primary or booster) 

           Thank you for this suggestion. We agree that comparing reasons for primary vs. booster hesitancy would enhance our findings, and help explain the associations observed in our models. Unfortunately, respondents who were primary hesitant could select reasons from a set of categories, where as booster hesitant respondents could only report reasons via free-text responses that were sparsely populated. We discuss this issue on page 8, lines 250-254, and hope this limitation can inform future data collection efforts.

3) the literature in this topic is wide, authors should include and discuss a larger part of it

            We have incorporated 6 additional, relevant citations into the manuscript.

4) the study is based on the follow-up of a previous cohort, and only subjects who had given their consent to be contacted, have been recruited. This can constitute a selection bias and should be discussed

            Thank you for raising this concern. We now discuss this limitation with our sampling strategy on page 8, lines 255 – 259:   

“Finally, we used the second round of MCAS for our data analysis, which only sampled respondents who participated in the first wave of the study. By only sampling from people that consented to a second wave of data collection, we may be introducing selection bias that makes our sample less representative of Minnesota as a whole.”

Minor concerns:

5) a more direct comparison with favors associated with vaccine hesitancy other US states with different demographic and social characteristics would be important to highlight the findings of the study      

            We no comment on state to state variation on page 7, lines 199 – 202.

            “A recent literature review found that vaccine hesitancy varied from around 10% to almost 90% within the United States by individual states and demographic groups, and our findings suggest Minnesota has lower hesitancy rates than many other states.”

6) Data presented at lines 175-177 are not represented in any figure

            We have updated this sentence to make it clear that these estimates represent sub-categories within the safety domain presented in figure 1 on page 6, lines 184 – 187.

7) no description of living conditions /e.g. rural/urban) is reported

               We now mention the lack of geographic and community measures in our data in the limitations section, page 8, lines 245-247

Round 2

Reviewer 2 Report

authors have addressed my comments